# The Evolution of the Satratoxin and Atranone Gene Clusters of *Stachybotrys chartarum*

**DOI:** 10.3390/jof8040340

**Published:** 2022-03-24

**Authors:** Sebastian Ulrich, Katharina Lang, Ludwig Niessen, Christiane Baschien, Robert Kosicki, Magdalena Twarużek, Reinhard K. Straubinger, Frank Ebel

**Affiliations:** 1Chair of Bacteriology and Mycology, Department of Veterinary Sciences, Faculty of Veterinary Medicine, Institute for Infectious Diseases and Zoonosis, LMU-Ludwig-Maximilians-University Munich, Veterinaerstr. 13, 80539 Munich, Germany; katharina.lang@micro.vetmed.uni-muenchen.de (K.L.); r.straubinger@lmu.de (R.K.S.); frank.ebel@micro.vetmed.uni-muenchen.de (F.E.); 2Chair of Microbiology, TUM School of Life Sciences, Technical University of Munich, Gregor-Mendel-Str. 4, 85354 Freising, Germany; ludwig.niessen@tum.de; 3Leibniz-Institute DSMZ-German Collection of Microorganisms and Cell Cultures, Inhoffenstr. 7B, 38124 Braunschweig, Germany; christiane.baschien@dsmz.de; 4Department of Physiology and Toxicology, Faculty of Biological Sciences, Kazimierz Wielki University, Chodkiewicza 30, 85-064 Bydgoszcz, Poland; robkos@ukw.edu.pl (R.K.); twarmag@ukw.edu.pl (M.T.)

**Keywords:** *Stachybotrys*, genome, macrocyclic trichothecene, atranone

## Abstract

*Stachybotrys chartarum* is frequently isolated from damp building materials or improperly stored animal forage. Human and animal exposure to the secondary metabolites of this mold is linked to severe health effects. The mutually exclusive production of either satratoxins or atranones defines the chemotypes A and S. Based upon the genes (satratoxin cluster, SC1-3, *sat* or atranone cluster, AC1, *atr*) that are supposed to be essential for satratoxin and atranone production, *S. chartarum* can furthermore be divided into three genotypes: the S-type possessing all *sat*- but no *atr*-genes, the A-type lacking the *sat*- but harboring all *atr*-genes, and the H-type having only certain *sat-* and all *atr-*genes. We analyzed the above-mentioned gene clusters and their flanking regions to shed light on the evolutionary relationship. Furthermore, we performed a deep re-sequencing and LC-MS/MS (Liquid chromatography–mass spectrometry) analysis. We propose a first model for the evolution of the *S. chartarum* genotypes. We assume that genotype H represents the most ancient form. A loss of the AC1 and the concomitant acquisition of the SC2 led to the emergence of the genotype S. According to our model, the genotype H also developed towards genotype A, a process that was accompanied by a loss of SC1 and SC3.

## 1. Introduction

The family of *Stachybotriaceae* has been described as genetically hyperdiverse and heterogeneous concerning the different secondary metabolite classes they produce (e.g., atranones, phenylspirodrimanes, and trichothecenes) [1,2]. According to the MycoBank database [3,4], the genus *Stachybotrys* currently consists of 126 species. Up to now, only two of them (*S. chartarum* and *S. chlorohalonata*) are frequently isolated and have been associated with illnesses affecting humans and animals [1,5,6]. *S. chartarum* occurs ubiquitously and is frequently found on dead plants (e.g., straw) and other cellulosic (e.g., culinary herbs) and building materials (e.g., gypsum and wallpaper) [2,7,8,9]. *S. chartarum* is either subdivided into two distinct chemotypes (A and S) based on their production of either atranones (chemotype A) or macrocyclic trichothecenes (MT; chemotype S) [10] or into the genotypes A, S, and H according to the presence or absence of genes that are presumed to encode the relevant enzymes for the biosynthesis of these mycotoxins (*atr*1-14 and *sat*1-21) [11]. The highly cytotoxic chemotype S was found to produce MT, whereas cultures of the low-cytotoxic chemotype A produce atranones [10,12,13,14]. MT represent the most cell-toxic trichothecenes currently known and include roridin, verrucarin, and satratoxins [15].

By binding irreversibly to the 60S ribosomal subunit, satratoxins inhibit protein biosynthesis and can induce apoptosis in neuronal cell lines [16,17,18,19]. The genotype S has been implicated in several types of diseases [20,21,22,23,24,25,26]: in animals, stachybotryotoxicosis can occur after oral uptake of mycotoxins, especially in horses and less frequently in cattle and sheep [27,28,29]. Humans, especially infants, are primarily at risk after exposure to toxins in water-damaged buildings [20,30]. Exposure to MT may cause pulmonary hemorrhage in infants or symptoms related to the sick building syndrome complex [20,31,32]. Data on the biological activity of atranones are still scarce, but the available information suggest that they are less cytotoxic than MT [5]. Rand et al. [33] showed that relatively high concentrations of atranone A and C (2.0–20 μg/animal) can cause an inflammatory response in mice, but clinical case reports for humans have not been published yet. Semeiks et al. [34] described the satratoxin cluster 1–3 (SC 1–3) harboring the *sat* genes 1 to 21 to be essential for satratoxin production and the atranone core cluster (CAC or AC1) that contains the *atr* genes (*atr*1–14) and may suffice for the production of atranones. The AC2 was described as a second atranone-specific gene cluster, but Semeiks et al. [34] already noted that three of the six genes are also conserved in genotype S strains. The function of this gene cluster is still unclear.

Based upon recent genetic information, three genotypes of the fungus have been introduced: genotype A produces atranones but no satratoxins and possesses *atr* but no *sat* genes; genotype H (hybrid) produces no satratoxins and has a truncated cluster of *sat* genes and all *atr* genes; and genotype S produces satratoxins and has the complete set of *sat* genes but no *atr* genes [5]. Remarkably, no *S. chartarum* strains have been isolated to date that lack the *atr* and the *sat* genes. The three genotypes have common but also distinct features, and their evolutionary relationship is unknown. Currently, genomes of two satratoxin-producing strains (IBT 40293 and 7711) but only one genotype A strain (IBT 40288) are available in the NCBI (National Center for Biotechnology Information) database (ASM102136v1 - GCA_001021365.1 - strain 51-11, S40293v1 - GCA_000732565.1 - strain IBT 40293, S7711v1 - GCA_000730325.1 - strain IBT 7711). The fourth *S. chartarum* genome that is available belongs to strain 51-11, for which no conclusive information exists so far concerning its chemotype. Up to now, no genome is available for a genotype H strain. Any analysis of the *S. chartarum* genomes is hampered by the fact that they are not fully assembled. Only sets of genomic scaffolds are deposited in the database and gene annotations are available only for three of the four genomes (IBT 7711, IBT 40293, and IBT 40288).

In this study, we analyzed the scaffolds harboring the satratoxin gene clusters SC1-3 and the atranone gene clusters AC1 and 2 (Appendix A). Additionally, we performed a deep re-sequencing of these gene clusters and an LC-MS/MS analysis of the atranone production using *S. chartarum* strains IBT 40293 and IBT 40288. The evolutionary process that led to the different genotypes in *S. chartarum* is unclear, and this was the primary motivation to perform the current study. Our data provide several clues for a better understanding of the evolutionary processes that shaped the present genotypes and their gene clusters and suggest *S. chartarum* as a promising model organism for further research on the evolution of mycotoxin-specific gene clusters.

## 2. Materials and Methods

### 2.1. Fungal Cultures and Culture Conditions

Based on previous data [11], we selected 1 out of 27 *S. chartarum* strains (IBT 40293, genotype S) for a deep re-sequencing analysis. Five strains of the S- and A-type (S-type: ATCC 34916, DSM 12880, DSM 2144; A-type: MS1, CBS 129.13) were selected to check by PCR whether particular sequence patterns are conserved. These strains were obtained from IBT (Institute for Bioteknologi, Lyngby, Denmark), CBS (Westerdijk Fungal Biodiversity Institute, Utrecht, The Netherlands), and DSMZ (Leibniz-Institute DSMZ-German Collection of Microorganisms and Cell Cultures, Braunschweig, Germany). A *sat*19-, *atr*6-, and *atr*4-specific triplex PCR was conducted according to Ulrich et al. [11] to confirm the genotype of each strain once a growing culture was obtained. Stocks were prepared and maintained in glycerol at −80 °C as described by Niessen and Vogel [35]. For DNA extraction, fungal material was transferred from potato dextrose agar plates (PDA) into liquid PD medium and cultured for 21 days at 25 °C and a_w_ 0.89 in the dark. For LC-MS/MS analysis, the strains were grown on PDA for 21 days at 25 °C and a_w_ 0.89 in the dark.

### 2.2. Sequence Analysis

The following genome assemblies that were available through NCBI were used for genome comparisons: *S. chartarum* S-type strains: IBT 40293 (GCA_000732565.1) and IBT 7711 (GCA_000730325.1) and the putative S-type strain 51-11 (GCA_001021365.1); A-type strain: IBT 40288 (GCA_000732765.1). The BlastN and BlastP tool was used to search for homologous sequences in the NCBI database [36,37]. Clustal Omega (EMBL, Hinxton Hall Conference Centre, UK) was used for alignments with default settings provided by the online platform. The alignment values were obtained using the MegAlignPro 17.1.1 software (DNASTAR, Madison, WI, USA) using MUSCLE (Multiple Sequence Comparison by Log-Expectation) with default settings provided by the software. Pairwise alignment was done using the Smith–Waterman algorithm provided by the previously mentioned software. Protein domains were predicted using the SMART algorithm (http://smart.embl-heidelberg.de/, accessed on 2 February 2022). Figures 1, 5, 6 and Appendix A were made using the BioRender© software (https://app.biorender.com/, accessed on 2 February 2022).

### 2.3. PCR

The PCR product downstream of SC1 was checked to confirm whether different genotypes A and S strains have a complete or truncated SC1-down-1 gene-gene. The PCR mastermix contained per 25 μL reaction 2.5 μL of 10x buffer, 0.5 μL of dNTP, 0.3 µL Taq DNA polymerase (each from Taq CORE Kit 10, MP Biomedicals, Eschwege, Germany), 0.5 μl of each primer (A-gen1-F: GCG GGC ACC AGG TGG GC; A-gen1-R: GAG TAC TCC ATC AAG TCC ATC CG; S-gen1-F: ACA TAT CCT CAT GCC TGC AGA; S-gen1-F: ATCGTCGAGTGATTCTGGAAGCG), 1 μL of DNA (final concentration 50 µg/mL), 0.25 µL of formamide (Sigma-Aldrich, Taufkirchen, Germany), and aqua dest. ad 25 μL. DNA amplification of the particular gene region was done using the following temperature protocol in a Mastercycler^®^ pro (Eppendorf AG, Hamburg, Germany): melting 1x 94 °C for 4 min followed by 35 × 94 °C for 60 s, annealing at 64 °C for 60 s, elongation at 74 °C for 60 s, followed by 1 × 72 °C for 4 min. The agarose gel contained 1% of agarose (Agarose Standard, Roti^®^garose, Biorad, Germany) and 4 μL GelRed^®^ Nucleic Acid Gel Stain (10.000X in water; Biotium, Fremont, CA, USA) per 100 mL TAE-buffer (1 x TRIS-Acetate-EDTA-Puffer, PanReac AppliChem, Darmstadt, Germany) and was operated for 90 minutes at 90 Volt. 1 μL of the sample was mixed with 4 μL aqua dest. and 1 μL 6x Loading Dye (ThermoFisher ScientificTM, Karlsruhe, Germany) before adding to the gel slot.

To verify our hypothesis that the scaffolds 1 (SC3 containing) and 1534 in the genome of IBT 40293 are neighboring, we designed the following oligonucleotides that bind in the boundary areas to bridge the putative gap: F-crossing-1 (AGT GCT GAT GGC AGG CGG CTA GCT CGA TCA) and R-crossing-1 (CCA ACA CCC TGG TCG GCT CTA TAG TCA CAT TG). The resulting fragment was sequenced at Eurofins Genomics (Ebersberg, Germany).

### 2.4. Deep Re-Sequencing Analysis

Deep re-sequencing analysis was conducted for strain IBT 40293 to compare the coverage of gene clusters implicated in the production of satratoxins (SC1-3) or atranones (AC1 and AC2), respectively. As described above, the strain was grown in liquid PD medium in a tissue culture flask. The mycelium (approximately 1.2 g/wet weight) was added to a 50 mL tube and centrifuged at 8228× *g*. The supernatant was discarded and the samples were washed two times with 20 mL sterile aqua. dest. Two hundred milligram of mycelium was dried in a 2 mL tube using a Thermomixer comfort (Eppendorf, Hamburg, Germany). Two glass beads (sterile 4 mm Ø) were added, and the sample was vortexed until the mycelium was pulverized. The beads were removed, and 1 mL cetyltrimethylammonium bromide (CTAB) was added. The sample was then incubated for 30 min at 65 °C and tilted every 10 min. After cooling on ice for 5 min, 1 mL chloroform was added, and the tube was tilted for 1 min. After centrifugation (10 min, 1088× *g*), the supernatant was transferred to a new vial, one volume of cold isopropanol (100%) was added, and the sample was carefully inverted. After another centrifugation (10 min, 1088× *g*, 4 °C), the isopropanol was discarded and replaced by 1 mL ice-cold ethanol (70%). After a final centrifugation step (10 min, 1088× *g*, 4 °C), the ethanol was removed with a pipette. The pellet was dried, and 100 µL buffer (10 mM Tris HCl, pH 7.5–8.5) was added. Deep re-sequencing and bioinformatic analysis were performed by Microsynth (Balgach, Switzerland). For coverage analysis, we used the NCBI database and, in particular, the following sequences: strain IBT 40293 (genotype S): SC1: accession number KL650302, SC2: KL651028, SC3: KL652499), and IBT 40288 (genotype A): AC1 (KL659150) and AC2 (KL656922).

### 2.5. Mass Spectrometry (LC-MS/MS)

To further verify previous information on the atranone production, both available strains (IBT 40293 and IBT 40288) were tested by mass spectrometry for their secondary metabolite profile [2,10]. Atranone extraction was conducted according to Andersen et al. [2] and the resulting samples were sent to the Department of Physiology and Toxicology (Kazimierz Wielki University, Faculty of Natural Sciences, Bydgoszcz, Poland). The following preparation steps were then performed according to the publication of Andersen et al. [10]. The samples were dissolved in a 1 mL mobile phase (A:B 4:1, see below) mixture. The extracts were then centrifuged for 30 min at 14,100× *g* and filtered using Millex^®^ 0.20 μm PTFE membrane filters (Merck, Darmstadt, Germany). Qualitative LC-MS/MS measurements were performed on a QTRAP 5500 tandem mass spectrometer (Sciex, Framingham, MA, USA) and a Shimadzu Nexera HPLC system equipped with a Kinetex C18 analytical column (100 × 2.1 mm, 2.6 µm; Phenomenex, Torrance, CA, USA). For the chromatographic determination of the analytes, mobile phase A (water + 0.1% acetic acid + 5 mM ammonium acetate) and mobile phase B (methanol + 0.1% acetic acid + 5 mM ammonium acetate) were applied using a gradient elution at a flow rate of 0.300 mL/min: 0 min 15% B; 14.2 min 75% B; 14.5 min 95% B; 17.0 min 95% B; 17.1 min 15% B; 22.0 min 15% B. The column temperature was set at 40 °C, and the injection volume of the samples was 10 μL. The mass spectrometry measurements were performed in the ESI-positive mode with an ion spray voltage of 4500 V and a source temperature of 550 °C. The curtain gas pressure was set at 20 psi, the heating gas and the nebulizer gas were both at 80 psi. The collision gas was operated in medium mode. All analytes were measured as adduct ions of hydrogen (M+H)+, and the toxins were identified in the multiple reaction monitoring mode (MRM). Table 1 shows the compound-specific parameters for the analytes of interest. Since there are currently no atranone standards available, we had to rely on the MRM provided by Andersen et al. [10]. MT were measured by LC-MS/MS in previous studies [11,38].

## 3. Results

Chemotype A and S strains are defined based on their production of atranones and satratoxins, respectively [10]. This is concomitant with the presence or absence of specific gene clusters (harboring the *atr* or *sat* genes) that define the A and the S genotype [11]. In the following text, we refer to individual strains as A- or S-type in accordance to their genotype. We analyzed the available genomic information of four *S. chartarum* strains, namely 51-11, IBT 40293, IBT 7711, and IBT 40288, to search for information that allows a better understanding of the genetic relationship of the different genotypes. The principal organization of the different clusters as described previously by Semeik, et al. [34] is summarized in Figure 1. An overview of the experiments performed in this study is given in Appendix A.

### 3.1. LC-MS/MS Analysis

From the four mentioned strains, we had access to only two: the S-type strain IBT 40293 and the A-type strain IBT40288. For strain IBT 40293, we recently described a profile of metabolites that is characteristic for an S-type strain [11]. For the A-type strain, IBT 40288, our current data confirm the production of atranones that are not produced by the S-type strain IBT 40293 (Table 2).

The other two strains, IBT 7711 and 51-11, were not accessible to us. However, data on their secondary metabolite profiles had been published. Andersen et al. [10] found that IBT 7711 produces MT but no atranones. For strain 51–11, toxicity was reported to be 192.0 T-2 equivalents/g (wet weight) after ten days at 23 °C on wet wallboard [39].

### 3.2. Deep Re-Sequencing Analysis

We also conducted a deep re-sequencing analysis for IBT 40293 to verify that the genomic information fits to the IBT 40293 strain that was present in our laboratory. As expected, a coverage analysis revealed that all genes of SC1–SC3 are present in this S-type strain (Figure 2A–C).

A comparison of the re-sequencing data from strain IBT 40293 with the genomic sequence of IBT 40288 demonstrated that the genes of the AC1 are lacking in IBT 40293 (Figure 3), but unexpectedly, we found that a short sequence corresponding to the edge of the AC1 is present in the IBT 40293 genome (Figure 3). This finding provides first evidence that the S-type strain IBT 40293 harbors a partial sequence of the *atr*1 gene. The up- and downstream sequences of the AC1 are very well conserved between IBT 40293 and IBT 40288, apart from two gaps downstream of AC1 that correspond to positions 70,513–71,026 and 71,417–73,403 of accession KL659150). One hypothetical protein (S40288_11253, position 71,829–73,238 bp) is predicted to be encoded in this region [33].

In contrast to AC1, the AC2 genes (2,926..14,620 bp of scaffold 123, accession KL656922) are also present in the S-type strain IBT 40293, whereas the downstream region of AC2 is also not well conserved between IBT 40293 and IBT 40288 (Figure 4).

### 3.3. G+C Content and Predicted Proteins

We focused our analysis on the SC and AC gene clusters and their up- and downstream regions (Figure 1, Figure 2, Figure 3, Figure 4, Figure 5 and Figure 6) and used the sequence information of the corresponding scaffolds (S-type strains: 51-11 (scaffolds 31, 111, 11, 4), IBT 40293 (scaffolds 155, 902, 1, 1203), and IBT 7711 (scaffolds 234, 1258, 1385, 1035); A-type strain: 40288 (scaffolds 1 and 123)).

The G+C content of the four available *S. chartarum* genomes is 53%. The G+C contents of the different clusters and the flanking genome sequences are summarized in Table 3A,B. All analyzed clusters and most up- or downstream regions have similar G+C contents (42–54%). Surprisingly, the up- and downstream parts of SC2 in strain 51-11 have a very low G+C content (27%). The corresponding regions in the other strains are difficult to compare since the sequences flanking the SC2 are only very short (approx. 772–886 bp).

The predicted Sat protein sequences are well conserved between IBT 7711 and IBT 40293 (97–100% identity). To obtain further information about the origin of the SC and AC genes, we conducted BlastP searches to identify orthologous proteins. Proteins that are homologous to Sat1-10 (SC1) and share a conserved domain structure were found in a variety of fungal species. Individual species harbor only a few or often just one of these homologous proteins. If several of these proteins are present in one species, the corresponding genes are scattered around the genome. An example are the *Stachybotrys elegans* proteins that are homologous to Sat1, Sat2, and Sat3 (IDs: KAH7316935.1, KAH7303828.1, and KAH7323111.1) (identity 70–88%, coverage 86–100%). A similar pattern was also found for proteins that are encoded by the SC2 genes, but the proteins with the highest homology and a conserved domain structure were identified in several *Monosporascus* species. Strikingly, the corresponding genes are organized in clusters. For *Monosporascus cannonballus,* one cluster comprises the neighboring genes encoding the homologs of Sat14 (RYO94656.1) and Sat15 (RYO94655.1); another cluster comprises ten putative trichodiensynthase genes, including those that encode the homologs of Sat11 (RYO75587.1), Sat12 (RYO75582.1), Sat13 (RYO75583.1), and Sat16 (RYO75586.1). This organization in two clusters is conserved in other genomes, e.g., *Monosporascus* sp. 5C6A.

BlastP searches were also performed for the SC3-encoded Sat17-21 proteins. Homologous sequences were found in several fungal species. Particular well-conserved proteins of all five SC3 proteins were identified in *Xylaria grammica.* In this endolithic fungus, four of the five genes are clustered (EKO27_g10813, EKO27_g10814, EKO27_g10815, and EKO27_g10816), and only the gene that encodes the Sat17-homologous protein (EKO27_g994) resides at a different position in the genome. In conclusion, individual genes encoding proteins with homology to certain Sat proteins are present in many fungal species. Gene clusters were only observed for homologs of SC2- and SC3-encoded proteins, e.g., in *Monosporascus* spp. and *Xylaria grammica.*

The AC1 cluster is also present in *S. chlorohalonata,* the second atranone-producing species of the genus *Stachybotrys* [2]. BlastP searches furthermore identified AC1 homologous proteins, often with a well-conserved domain structure, in many other fungal species. No evidence for a clustering of the corresponding genes was observed.

### 3.4. Satratoxin Cluster Alignment

#### 3.4.1. Satratoxin Cluster 1 (SC1)

The SC1 in the two S-type strains (IBT 7711 29,996 bp, IBT 40293 29,998 bp) and the atypical S-type strain (51-11: 28,909 bp) have a similar length and show a high level of sequence conservation (compare Table 4). The SC1 (according to [34]) comprises ten genes: *sat*1–10 (compare Figure 1A). All of them are present in strains IBT 7711 and IBT 40293, whereas strain 51-11 lacks the *sat*1 gene. All other gaps are within non-coding elements.

The available sequences upstream of SC1 are relatively short for 51-11 (1638 bp) compared to the other strains (IBT 7711: 4564 bp and IBT 40293: 4155 bp). The two latter regions are well conserved to each other (85.7% identity). Only one gene is annotated in this region for strain IBT 7711 (identifier S7711_11135).

The loss of *sat*1 in strain 51-11 prompted us to perform a detailed analysis of the 1638 bp upstream of *sat*2 present in scaffold 31 of strain 51-11. Remarkably, a short sequence of 230 bp is also present in the other strains (IBT 7711 (scaffold 518): identity 87%, coverage 100%; IBT 40293 (scaffold 506): identity 89%, coverage 97%) and covers the intergenic region between *sat*2 and *sat*1 (1373 bp, 99% identity) and the last 35 bp of the *sat*1 gene (97% identity) (Appendix A).

Approximately 10,000 bp downstream of SC1 are available for all three S-type strains, and these sequences are well conserved (identity: 98.7–99.3%, coverage 100%). Two hypothetical proteins are predicted immediately downstream of the cluster and were tentatively designated SC1-down-1 and SC1-down-2. We used BlastN to search for homologous sequences in other fungi and obtained only one hit for *Purpureocillium lilacinum* (isolate PLBJ-1 scaffold00012): the hypothetical proteins VFPBJ_11029 and VFPBJ_11030. These genes share 77% and 82% identity with SC1-down-1 and SC1-down-2, respectively. Both are neighboring genes and have the same convergent orientation in *S. chartarum* and the *P. lilacinum* strains PLBJ-1 and PLFJ-1. Sequence comparison of the surrounding region from both *P. lilacinum* strains revealed that the homology extends further to a sequence sharing 73% identity with *sat*10. The corresponding *P. lilacinum* protein (A0A179HJD5) is smaller than Sat10 (KEY74377) (1265aa versus 1930aa), but both proteins contain equal numbers of ankyrin repeats (*n* = 16).

The data presented above suggest a common site of insertion for the SC1 in the three S-type strains since two upstream and all three downstream sequences are well conserved. The only divergent region, the upstream region in strain 51-11, had apparently been reorganized later on. We also used the sequences up- and downstream of SC1 from the three S-type strains to identify and compare the corresponding regions in the only available A-type genome (IBT 40288). We identified a highly homologous region to the sequences immediately downstream of SC1 (scaffold 1204 of IBT 40288) comprising a complete SC1-down-2 gene (100% identity) but only a partial sequence of SC1-down-1 (241 bp of 3692 bp). We used a PCR approach to analyze other S- and A-type strains (S-type: ATCC 34916, DSM 12880, DSM 2144; A-type: MS1, CBS 129.13) for the presence of a full-length or truncated SC1-down-2 gene (Appendix A). All S-type strains have a full-length SC1-down-1 gene indicated by a long and a short PCR product, whereas all A-type strains lack the larger PCR product.

No homologous sequence was found in IBT 40288 corresponding to the region upstream of SC1. We also used the 4910 bp sequence upstream of the truncated SC1-down-1 gene in IBT 40288 to search for homologous sequences in the available *S. chartarum* S-type genomes but found no hit. This suggests that this sequence, which contains two hypothetical proteins (identifiers S40288_06821 and S40288_11170), is specific for A-type strains.

#### 3.4.2. Satratoxin Cluster 2 (SC2)

We also aligned the sequences of the SC2 gene cluster-containing scaffolds of the three S-type strains (51-11: scaffold 111, IBT 40293: scaffold 902, 7711: scaffold 1258). The corresponding results are summarized in Table 5 and Figure 1B.

The predicted exons encoding the *sat*11–16 genes are well conserved, and the alignments of the SC2 sequences show that the sequences of 51-11, IBT 40293, and IBT 7711 are highly homologous (99.5–99.6%). The lengths of the SC2 were also very similar (51-11: 19,876 bp, IBT 7711: 19,854 bp, and 40293: 19,873 bp). The available upstream sequences are relatively short, with a minimum of 772 bp for strain IBT 7711 and a maximum of 2133 bp for strain 51-11. These sequences show high similarity between 51-11, IBT 40293, and IBT 7711 (99.0–99.1%). The available sequences downstream of SC2 differ in length for the three strains (51-11: 8737 bp, IBT 7711 bp: 2.775, 40293: 2768 bp) but are also well conserved, indicating that the SC2 is inserted at the same position in all three strains. No genes are currently annotated up- or downstream of the SC2. Searching the IBT 40288 genome for sequences that are homologous to the SC2-flanking regions resulted in no unambiguous hit.

#### 3.4.3. The Satratoxin Cluster 3 (SC3)

In IBT 7711 and IBT 40293, the SC3 is positioned next to the trichodiensynthase cluster (CTC). We therefore presumed to find the SC3 genes of strain 51-11 at the same position, but a BlastN search revealed homologous sequences neither downstream of the CTC nor in any other scaffold. This finding indicates that 51-11 lacks the SC3. An alignment of the sequences from IBT 7711 and IBT 40293 showed that the genes of the SC3 (*sat*17–21) are all present and well-conserved (identities > 99.1%) (Figure 1C). The upstream region of SC3, harboring the CTC, is also very similar (>99%) and contains all trichodiensynthase genes. The different gene order of the CTC genes compared to other trichothecene-producing strains (e.g., *Fusarium sporotrichioides*) is conserved in all A- and S-type strains. The available downstream sequences are extremely short (IBT 7711: 122bp and IBT 40293: 112bp) and well conserved.

Aligning the SC3 containing scaffolds of the S-type strains with the available sequences of IBT 40288 revealed no significant homologies apart from the trichodiensynthase genes. The sequences downstream of the CTC in strains 51-11 and IBT 40288 are well conserved for the coding and non-coding sequences (identities 100%; length IBT 40288: 59,315 bp, 51-11: 273,615 bp).

The putative genes of strains 51-11 and IBT 40288 that reside at the expected position of the SC3 are also present in the genomes of IBT 7711 and IBT 40293 (identity 99%) but not in the SC3-containing scaffold. These regions and the two SC3 sequences each reside at the edges of their scaffolds in IBT 7711 and IBT 40293, which prompted us to investigate by PCR whether both scaffolds are in fact continuous. We indeed obtained a PCR product for IBT 40293, and sequencing revealed that both scaffolds are separated by only 177 bp (Appendix A). Thus, the SC3 resides at the same position in the genomes of IBT 40293 and (most probably) IBT 7711, but it is lacking in strain 51-11.

### 3.5. Atranone Cluster 1 and 2

#### 3.5.1. Atranone Cluster 1 (AC1)

According to our BlastN searches, none of the *atr* genes (of AC1) is present in the three S-type strains. However, 170 bp comprising parts of the putative promoter region of *atr*1 and a truncated *atr*1 coding sequence (*atr*1: 242 bp, 80.4% identity, coverage 42%) were found in IBT 40293 and 51-11 but not in IBT 7711 (Appendix A and Figure 5). This finding is in line with the re-sequencing data presented in Figure 3, in which the *atr*1 gene is positioned at the left margin of the cluster. In strains IBT 40293 and 51-11, sequence information downstream of this region is unavailable since this region represents the end of the respective scaffolds 1474 and 47. The presence of a short but well-conserved part of the *atr*1 gene in two S-type strains is remarkable and suggests that the ancestor of these strains harbored an *atr*1 gene.

**Figure 5 jof-08-00340-f005:**
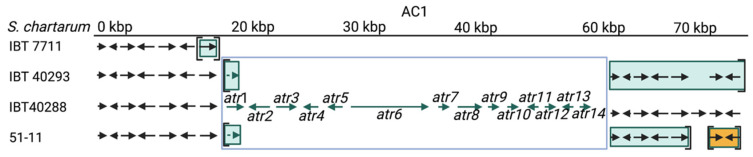
The atranone cluster 1 (AC1) region annotated for four *S. chartarum* strains. Arrows indicate genes. The boxed genes represent the AC1. Genes in square brackets are not part of the scaffold harboring AC1 and/or its upstream region but reside instead in other scaffolds. (IBT 7711: upstream = scaffolds 1183 and (1150)), IBT 40293: upstream = scaffold 1452; downstream = (scaffold 1474)), IBT 40288: scaffold 1 and 51-11: upstream = scaffold 47; downstream are five genes of scaffold (127) and two genes of scaffold (11)). An identical coloration indicates affiliation to a common scaffold in the respective strain.

The available sequence upstream of AC1 (in IBT 40288) is well conserved in all other *S. chartarum* genome sequences (identity 99%). In strain IBT 7711, the upstream part is divided into two scaffolds (scaffolds 1183 and 1150), but these may be neighboring. This is suggested by the continuous sequences upstream of AC1 in strains IBT 40293 and IBT 40288 and the fact that the two IBT 7711 sequences are located at the edges of their scaffolds, but an attempt to bridge the potential gap by PCR was so far not successful (data not shown). The region downstream of AC1 is also very well conserved except for strain IBT 7711. The only remarkable difference between the A-type and the two S-type strains is that one hypothetical protein (identifier S40288_11253) seems to be A-type-specific (compare Figure 5). In strain 51-11, the downstream region is allocated in two different scaffolds. The respective genes are not found at the edges of their scaffolds, indicating that they are not positioned next to each other but reside in different parts of the 51-11 genome. The position where the two regions of 51-11 have putatively been separated is precisely the position at which the genotype A-specific gene S40288_11253 is found in IBT 40288 (Figure 5).

#### 3.5.2. Atranone Cluster 2 (AC2)

The AC2 was defined by Semeiks et al. [34] and comprises six genes; five of them are well conserved in the three S-type strains (approx. 88% identity and 99% coverage) (Figure 6 and Table 6), indicating that AC2 is not A-type specific. The region downstream of AC2 apparently underwent significant reorganizations (identity 54.1–74.5%, coverage 64.4–86.0%). These putative genes are mostly well preserved in all three S-type strains but are scattered over several scaffolds (compare Figure 6). No sequence information is currently available for the region upstream of AC2.

An analysis of the putative protein sequences of AC2 revealed the Pho4 domain (InterPro entry IPR001204) in protein S40288_09036 and the helix-loop-helix domain in S40288_11459 that were previously described by Semeiks et al. [34]. We furthermore found five transmembrane regions in S40288_11460 and an LMWPc domain (pfam01451) commonly present in low-molecular-weight phosphatases in S40288_09035. The latter protein is the one that exists only in the A-type strain IBT 40288 (compare Figure 5).

**Figure 6 jof-08-00340-f006:**
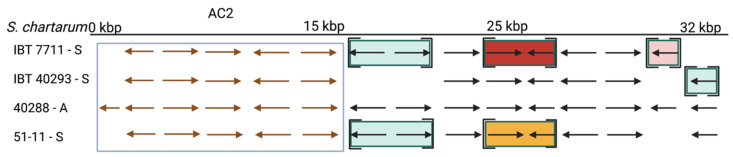
The atranone cluster 2 (AC2) region annotated for *S. chartarum* strains IBT 7711, IBT 40293, IBT 40288, and 51-11. Arrows indicate genes. The boxed genes represent the AC2 and encode the hypothetical proteins: S40288_09035, S40288_11459, S40288_09036, S40288_09037, S40288_09038, and S40288_11460 (from left to right). Genes in square brackets are not part of the scaffold harboring the AC2 but reside in other scaffolds. An identical coloration indicates affiliation to a common scaffold in the respective strain that is distinct from that harboring the AC2.

## 4. Discussion

Mycotoxins are produced by distinct and elaborated biosynthetic pathways that have often been elucidated in only one species [40]. The class of trichothecenes is particular variable in its chemical structure and has therefore been divided, based on functional groups, in the types A to D [18]. MTs are also known as type D trichothecenes; they are complex molecules with particularly high toxicity compared to the simpler trichothecenes of the types A to C [18].

Certain MTs are also produced by different fungi other than *S. chartarum*, but within the species, they are exclusively produced by genotype S strains. Atranones belong to a different mycotoxin class and are generated by genotype A strains of *S. chartarum* as well as other *Stachybotrys* species, e.g., *S. chlorohalonata*. Currently, *S. chartarum* is not included in the MIBiG (Minimum Information about Biosynthetic Gene cluster) [41] since a functional validation for the SC and AC gene clusters is still pending. MTs are the putative end products of the trichothecene biosynthesis pathway that is still elusive, but several possible mechanisms have been proposed in the literature [18,34,42].

Biosynthetic gene clusters are highly variable, rapidly evolving, and often found on accessory chromosomes or at the ends of chromosomes [43,44,45]. The available genomes of *S. chartarum* are not fully annotated and exist as sets of genomic scaffolds. Hence, we know the precise position of an AC or SC within the respective scaffold but not on the corresponding chromosome. Most of these clusters are apparently not residing at the very end of a chromosome. This is less clear for AC1, for which only a short downstream sequence is present in the database.

The genome of *S. chartarum* 51-11 comprises the most continuous sequences and thereby provides a framework to search for connections between individual scaffolds in the other *S. chartarum* genomes. Horizontal gene transfer may be involved in the evolution of the different *S. chartarum* genotypes, and this can result in a divergent G+C content [46]. We therefore analyzed the G+C content of the gene clusters and the flanking regions of the respective scaffolds. All gene clusters have a similar G+C content as the rest of the genome, suggesting that they originate from *Stachybotrys* or a fungus with similar G+C content.

The genotype S is defined by the presence of the SC1-3 and the absence of the AC1 [34], criteria which are met by IBT 40293 and IBT 7711. No annotation is available for the genome sequence of strain 51-11. In this study, we have identified an SC1 and SC2 but no AC1 or SC3; hence, 51-11 does not fully match the criteria of a genotype S strain, and we therefore refer to it as an atypical S-type strain. The analysis of the flanking regions of the SC1-3 clusters indicates that their chromosomal positions are identical or at least very similar in the three S-type strains, which corroborates preliminary findings [11]. From these data, it appears unlikely that the SCs are very mobile genetic elements in the *S. chartarum* genome.

The situation is different for the AC. The AC1 cluster is present in the A-type strain IBT 40288 and H-type but not in S-type strains [11,34]. AC2 is not an A-type-specific region since it is present in all A- and S-type genomes analyzed in this study. According to our data, it is unlikely that the AC2-encoded proteins are directly involved in the production of atranones. No information is currently available on the presence and position of AC2 in H-type strains. In the A- and S-type genomes, AC2 resides in different positions, indicating that these clusters were acquired in several distinct events or that the respective regions underwent substantial genomic rearrangements.

A central question arising from the different genotypes of *S. chartarum* is their evolutionary relationship. We have obtained several hints that shed light on the relations of the different genotypes and encouraged us to develop a model that is consistent with our findings.

For unknown reasons, all *S. chartarum* strains we know of possess either an AC1 and/or specific satratoxin gene clusters. This is true for the A- and S-type but also for H-type strains [11]. In a study comprising 105 *S. chartarum* strains, none possessed or lacked both the AC1 and the SC2 [47]. This strongly suggests that AC1 and SC2 are mutually exclusive. It is remarkable in this context that we have no evidence that *S. chartarum* strains exist in which an intact SC2 coexists with an incomplete AC1 or vice versa. This finding is striking and suggests that acquisition of either of these gene clusters always entails a complete excision of the other. Although the precise pathway for the biosynthesis of MTs or atranones is unknown, we know that farnesylpyrophosphate is the starting compound for both processes [14,18]. It appears that each strain has to choose whether it converts farnesylpyrophosphate into the direction of atranones or satratoxins. This is also supported by the genome of strain 51-11. Unfortunately, this strain is currently unavailable for culturing and MT analysis. The published data on the toxicity of 51-11 provide only hints but no concrete information on its MT production [39]. Thus, it is unclear whether strain 51-11 is a typical chemotype S strain. It would be interesting to know whether this strain produces and releases satratoxins even though the supposed MFS transporter (encoded in the SC3) is missing. Strain 51-11 differs from the other S-type strains since it lacks not only the SC3 cluster but also most of the *sat*1 gene of the SC1. This apparent loss of genetic information indicates that 55-11 developed from a typical S-type strain (Figure 7).

If AC1 and SC2 are mutually exclusive, it is conceivable that an ancestor harbored one of these clusters, but the question remains: which one was first? The genotype H may provide clues for a better understanding of the evolutionary processes that shaped the current genotypes since it combines elements present in A- or S-type strains. According to the hypothesis of Semeiks et al. [34], the SC2 developed from a duplication of the CTC, which suggests that the S-type emerged from an A- or H-type ancestor. The fact that we found truncated *atr*1 sequences in all S-type strains strengthens this hypothesis. It is, in principle, possible that the duplication of the CTC took place in an ancestral strain that contained only the CTC cluster, but such a strain has not been identified yet. Following the duplication of the CTC, a functional divergence may have resulted in the emergence of the SC2 and the parallel loss of the AC1. In this context, it is remarkable that BlastP searches identified a gene cluster in *Monosporascus* species, which encodes proteins that are homologous to specific SC2-encoded proteins but also trichodiensynthases that are homologous to proteins, which are encoded in the CTC of *S. chartarum* (Appendix A). Since *Monosporascus* is an important spoilage agent in melons [48] and was recently presumed to produce MT (roridin E) [49], this finding may also be relevant with respect to potential adverse health effects to humans.

Our analyses also showed that the region located immediately downstream of the SC1 is particularly interesting. It is striking that the SC1-down-1 gene is well conserved in all S-type strains but truncated in the A-type strain and that the lost part of the gene would be most proximal to the SC1. We assume that the SC1-down-1 gene was destroyed by an imprecise excision of the SC1, which implies that the A-type developed from a strain harboring the SC1, hence an H- or S-type strain.

The A-type strain IBT 40288 possesses a complete AC1 cluster, and our previously published PCR data suggest the same for all H-type strains analyzed so far [11]. Further data strongly indicate that H-type strains are genetically heterogeneous and often lack one or more of the SC1 and/or SC3 genes [11,47]. This loss of genetic information can be construed as an erosive process, in which H-type strains derive from a strain harboring intact SC1 and SC3 and developed towards strains that lack these genetic elements. To integrate this into our model, we postulate the existence of an ancestral H-type variant with intact and functional SC1 and SC3 clusters that is represented by strain CBS 324.65.

Based on the currently available genomic information (summarized in Figure 1, Figure 5 and Figure 6), we propose the model that is depicted in Figure 7; it suggests that the H-type is the ancestor rather than the successor of the A-type. The H-type comprises a heterogeneous group of strains that have in common the presence of the SC1 and SC3 and the absence of the SC2 but differ with respect to the genetic erosion of the SC1 and SC3. Our model takes four findings into account: (1) the presence of truncated *atr*1 sequences in the S-type strains (Appendix A), (2) the presence of incomplete SC3 clusters in H-type strains, (3) the truncated SC1-down-1 gene in the A-type strain, and (4) the loss of the SC3 and the *sat*1 gene in the atypical S-type strain 51-11. Our model implies that H-type strains represent a crucial element for a better understanding of the evolution of the *S. chartarum* genotypes.

Another essential question that arises from the current study is how the different gene clusters collaborate to facilitate the production of mycotoxins, e.g., satratoxins and atranones. Usually, gene clusters contain so-called backbone biosynthesis genes for the production of individual mycotoxins. Genes that are engaged in the same secondary pathway commonly form gene clusters that are stable and rarely divided [50,51]. In *S. chartarum*, three clusters have been implicated in the biosynthesis of satratoxins despite the fact that they are spread in the genome. This situation is similar for the gene clusters that enable the synthesis of the T-2 toxin in *Fusarium sporotrichioides* [52,53], which is chemically closely related to the satratoxins. A physical separation of gene clusters suggests that they have different functions but does not exclude synergistic interactions. All three SC harbor one putative transcription factor (SC1: Sat9, SC2: Sat15, and SC3: Sat20). This pattern is often found for secondary metabolite clusters and allows a separate regulation for each set of genes. Apart from the biosynthesis, mycotoxins have to be released into the environment. Major Facilitator Superfamily (MFS) transporters have been implicated in this process [54]. It is remarkable that the only MFS protein encoded by the *sat* genes is Sat21, which is encoded in the SC3. Semeiks et al. [34] therefore assumed that the SC3 plays an important role in the transport of trichothecenes, a notion that is further supported by the fact that the SC3 and the CTC reside next to each other. The fact that the atranone-producing strains either lack a SC3 cluster (A-type) or often lack the *sat*21 gene (certain H-type strains) [47], indicating that atranones are most likely exported via a mechanism that does not involve Sat21. How the proteins that are encoded by the SC clusters interact and cooperate is still an open question. Whether the mycotoxin biosynthesis takes place in the cytoplasm or whether certain steps are concentrated in certain intracellular compartments or organelles is another. Sat10 is a further noticeable SC-encoded protein. It comprises 16 ankyrin repeat domains and is therefore supposed to be a scaffolding protein that interacts with other Sat proteins to coordinate their activities. The *sat*10 gene is the last gene of the SC1 cluster. Using BlastN, we found that the two *S. chartarum* genes downstream of the SC1 are well conserved in the fungus *Purpureocillium lilacinum.* Strikingly, a neighboring gene shows high homology to *sat*10 and also encodes a protein with 16 ankyrin repeat proteins. The fact that these three genes are well conserved both with respect to their sequence and their orientation strongly suggests that they form a genetic element that was transferred from *S. chartarum* to *P. lilacinum* or vice versa. Such a horizontal gene transfer has been described for other fungal genes, e.g., for those involved in the sterigmatocystin synthesis in *Aspergillus* species [55,56]. Similar mechanism may have also enabled the transfer of MT-related genes to or from *Myrothecium* species. The fact that the plant *Baccharis mesapotamica* is able to produce roridins and verrucarins even suggests the possibility of an inter-kingdom transfer of genetic information [57,58].

Three distinct mechanisms can drive the evolution of functionally diverse gene clusters: de novo assembly, functional divergence, and horizontal transfer [43]. De novo assembly is the most time-consuming and therefore difficult step; it is required to generate an initial gene cluster that can then develop further. Our BlastP searches identified proteins with substantial homology to Sat or ATR proteins in a variety of fungal species, and in most cases, the domains present in the Sat or Atr proteins are also well conserved. Remarkably, only few species harbored several of these genes in their genome. *S. elegans* is one example, and its genome contains three genes encoding proteins that are homologous to Sat1, Sat2, and Sat3, but strikingly, these genes are not clustered. Hence, a pool of fungal genes exist that is scattered over many species and likely provided the raw material for the assembly of gene clusters, such as those comprising the *sat* and *atr* genes. It is likely that the assembly of such clusters occurs gradually starting with small clusters that are successively enlarged by additional genes. The identification of the gene cluster in *Monosporascus* spp. and *Xylaria grammica* indicate that gene assemblies that are related to SC2 and SC3 exist in other fungi. A closer analysis and comparison of these clusters may provide further important insights. Functional divergence could explain how the SC2 emerged from a duplication of the CTC. Alternatively, the CTC and SC2 may derive from a common ancestor, and the *Monosporascus* gene cluster in fact appears to be a chimera of CTC and SC2. The presence of similar gene clusters in *S. chartarum* and other fungi strongly suggest that horizontal gene transfer played an important role in the evolution of the *sat* and *atr* gene clusters.

## 5. Conclusions

*S. chartarum* is an attractive and promising model organism to study biosynthetic pathways that allow the production of different sets of mycotoxins within one species due to the existence of at least three distinct genotypes, which determine the capacity of individual strains to produce certain mycotoxins. *S. chartarum* is furthermore an ideal model to analyze evolutionary processes that shaped the current gene clusters and also to study their functional interplay, e.g., in the production of satratoxins. The results of the current study demonstrate that the mining of genomic information can enable us to reconstruct key steps in the development of the current genotype. Our data suggest that both the A- and the S-type emerged from an ancestor that resembled the current H-type. The H-type appears to be a key element in the evolution of the other genotypes and thereby a key factor for a better understanding of the underlying processes. H-type strains are defined by the presence of an AC1 and the absence of an SC2, but apart from these common features, they are also heterogeneous with respect to the completeness of their SC1 and SC3 clusters. Therefore, genomic information on H-type strains will be vital to unravel the evolution of the *S. chartarum* gene clusters. The current study demonstrates that the careful search for relicts that arose from imprecise genetic reorganizations can provide valuable insights into evolutionary processes. This genomic archeology approach should be further pursued to unravel the entangled ways that led to the current architecture of fungal gene clusters.

## Figures and Tables

**Figure 1 jof-08-00340-f001:**
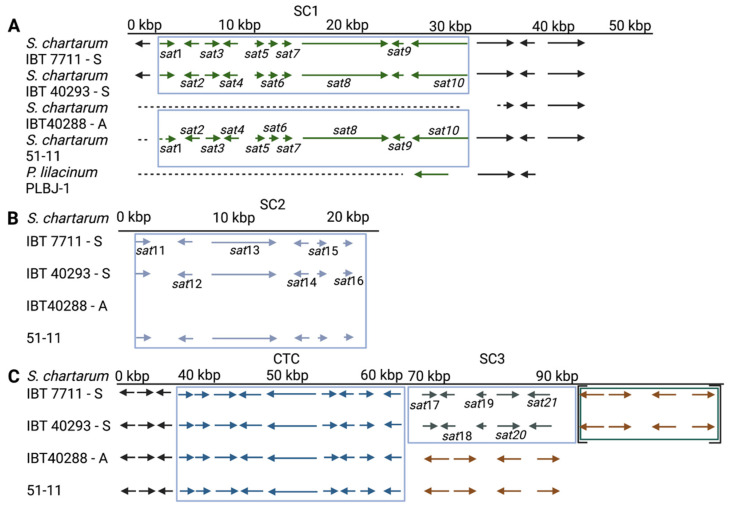
Satratoxin clusters 1–3 (SC1-3, (**A**–**C**)) and the core trichothecene cluster (CTC) of *S. chartarum* strains 51-11, IBT 40293, IBT 7711 (genotype S strains), and IBT 40288 (genotype A strain). Arrows depict genes; the different gene clusters are boxed. A dotted line indicates that a sequence is not existent in the respective genome. The genes in square brackets (Panel **C**) are annotated to a different scaffold than the other genes of these strains, but our PCR and sequencing results demonstrate for IBT 40293 that these regions are part of a continuous sequence. (Panel **A**) also shows a region of three homologous genes of the *Purpureocillium lilacinum* isolate PLBJ-1 for comparison. The brown arrows in (Panel **C**) indicate a set of four genes that are present in all four strains.

**Figure 2 jof-08-00340-f002:**
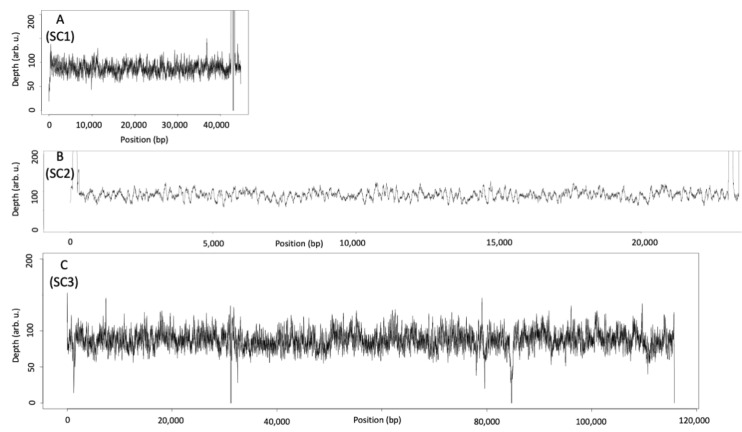
Coverage analysis of deep resequencing sequences of IBT 40293 with accession numbers KL650302, KL651028, and KL652499 (representing the (**A**) SC1, (**B**) SC2, and (**C**) SC3 of IBT 40293, respectively).

**Figure 3 jof-08-00340-f003:**
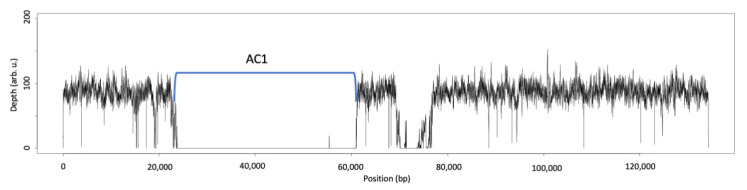
Coverage of accession KL659150 representing the AC1 of IBT 40288 (A-type strain) by sequences derived from IBT 40293 (S-type strain). The AC1 is indicated and reaches from position 24,509 bp (*atr*1) to position 60,118 bp (*atr*14). Note that a short sequence at the left margin of the AC1 is covered by the IBT 40293 sequences.

**Figure 4 jof-08-00340-f004:**
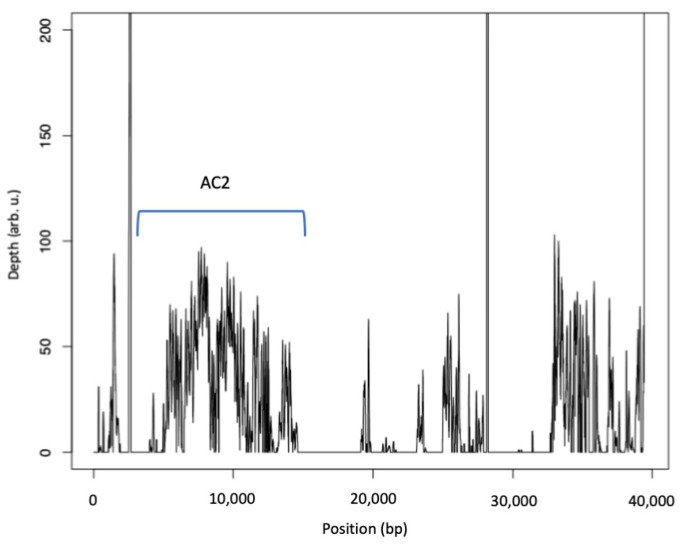
Coverage of accession KL656922 representing the AC2 of IBT 40288 (A-type strain) by sequences derived from IBT 40293 (S-type strain). The position of the AC2 is indicated.

**Figure 7 jof-08-00340-f007:**
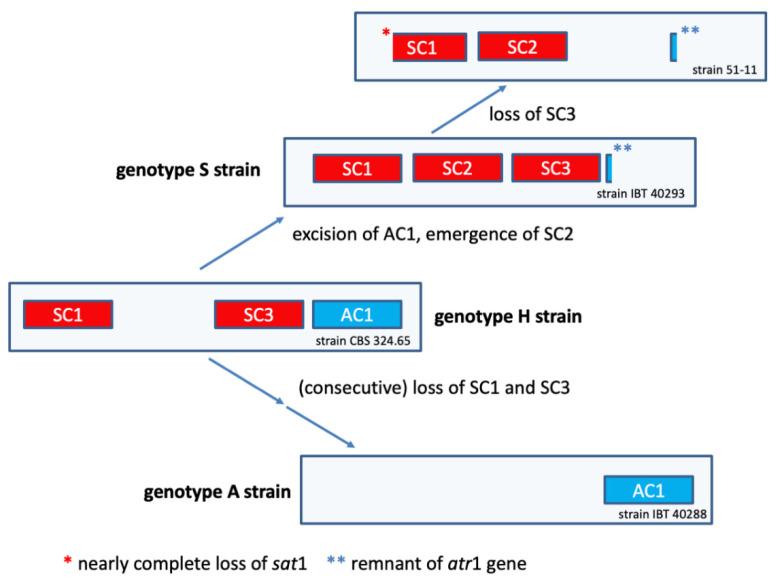
Current model of the evolution of the *S**. chartarum* genotypes S, A, and H. The different gene clusters are depicted as boxes. The size and the position of these boxes do not represent the correct size and position of the respective cluster in the genome. Satratoxin gene clusters are shown in red and atranone gene clusters in blue.

**Table 1 jof-08-00340-t001:** MRM-transitions and substance-specific parameters for the identification of 6-hydroxydolabella, 3,4-epoxy 6-hydroxydolabella, and atranone A by LC-MS/MS.

Compound	Q1 (*m/z*) ^a^	Q3 (*m/z*) ^a^	Retention Time (min) ^b^	DP (V)	CE (V)	CXP (V)
6-hydroxydolabella-3,7,12-trien-14-one	303.3	205.1	12.0	86	15	12
	303.3	243.0	12.0	86	11	14
3,4-epoxy 6-hydroxy-dolabella-7,12-diene-one	319.2	221.1	10.5	101	11	12
	319.2	259.0	10.5	101	9	14
atranone A	417.3	205.2	11.4	200	33	12
	417.3	357.2	11.4	200	15	11

^a^ qualitative measurement; ^b^ probable retention time; DP, declustering potential; CE, collision energy; CXP, cell exit potential.

**Table 2 jof-08-00340-t002:** Detected atranone precursors and atranone A by LC-MS/MS measurement.

Sample Name	6-hydroxydolabella-3,7,12-trien-14-one		3,4-epoxy-6-hydroxy-dolabella-7,12-diene-one		Atranone A	
	Quantifier	Qualifier	Quantifier	Qualifier	Quantifier	Qualifier
	303.2/205.1 Da	303.2/243.0 Da	319.2/221.1 Da	319.2/259.0 Da	417.3/357.2 Da	417.3/205.2 Da
IBT 40288	230,000 *	54,900 *	349,000 *	107,000 *	491,000 *	379,000 *
IBT 40293	n.d.	n.d.	n.d.	n.d.	n.d.	n.d.

n.d., not detected; * area of the detected peak.

**Table 3 jof-08-00340-t003:** (**A**,**B**): Comparison of the genomic G+C content of the SC and AC and their up- and downstream regions of strains 51-11, IBT 40293, IBT 7711, and IBT 40288. The GC content (in %) is given in normal-sized numbers; the length of the respective sequence is given in subscript. Bp, base pairs; n.d., not determined; - - -, missing. * fragments present in different scaffolds; ** sequence divided into two parts residing in different scaffolds.

**A: Satratoxin Clusters**									
**Strain ID**	**SC1 scaffolds**			**SC2 scaffolds**			**SC3 scaffolds**		
	**Upstream**	**SC1**	**Downstream**	**Upstream**	**SC2**	**Downstream**	**Upstream**	**SC3**	**Downstream**
51-11	45_1638bp_	53_29,998bp_	48_140,301bp_	27_2133bp_	47_19,876bp_	27_140,301bp_	51_849,200bp_	- - -	- - -
IBT 40293	44_4155bp_	53_28,909bp_	51_10,613bp_	42_886bp_	47_19,873bp_	44_10,613bp_	51_105,159bp_	53_10,500bp_	48_112bp/42,552bp_ **
IBT 7711	45_4564bp_	53_29,996_	50_72,574bp_	45_772bp_	47_19,854bp_	43_72,574bp_	51_95,549bp_	54_10,496bp_	33_122bp_
IBT 40288	- - -	- - -	52_96,283bp_	- - -	- - -	- - -	51_129,561bp_	- - -	- - -
**B** **: Atranone Clusters**									
**Strain ID**	**AC1 scaffolds**						**AC2 scaffolds**		
	**Upstream**		**AC1**	**Downstream**			**AC2**		**Downstream**
51-11	46_85,474bp_		- - -	39_21,564bp/849,200bp_ **			53_10,350bp_		53_n.d._ *
IBT 40293	49_42,470bp_		- - -	49_54,221bp_			57_9340bp_		54_294,349bp_ *
IBT 7711	50_33,211bp/12,367bp_		- - -	- - -			53_10,380bp_		53_64,167bp_ *
IBT 40288	45_24,509bp_		49_35,610bp_	48_74,142bp_			51_11,699bp_		51_24,996bp_

**Table 4 jof-08-00340-t004:** Comparison of the genomic regions that comprise the SC1 of 51-11, IBT 40293, and IBT 7711 (scaffolds 31, 155, 234, respectively).

Strain ID	Upstream of SC1		SC1 Genes (*sat*1-10)		Downstream of SC1	
	Identity (%)	Gaps (%)	Identity (%)	Gaps (%)	Identity (%)	Gaps (%)
51-11 vs. IBT 40293	46.5	37.6	95.7	4.1	98.7	0.6
51-11 vs. IBT 7711	45.4	40.0	95.7	4.1	98.5	0.6
IBT 40293 vs. IBT 7711	85.7	11.7	99.8	0.0	99.3	0.1

**Table 5 jof-08-00340-t005:** Comparison of the satratoxin cluster 2 (SC2)-containing scaffolds (51-11: scaffold 111, IBT 7711: 1258, IBT 40293: 902).

Strain ID	Upstream of SC2		SC2 Genes (*sat*11-16)		Downstream of SC2	
	Identity (%)	Gaps (%)	Identity (%)	Gaps (%)	Identity (%)	Gaps (%)
51-11 vs. IBT 40293	99.0	0.3	99.5	0.1	98.5	0.2
51-11 vs. IBT 7711	99.0	0.6	99.5	0.1	98.6	0.1
IBT 40293 vs. IBT 7711	99.1	0.8	99.6	0.1	98.8	0.2

**Table 6 jof-08-00340-t006:** Comparison of the presumed coding sequences of the atranone cluster 2 (AC2) and the respective scaffolds ((IBT 40288: scaffold 123); 51-11: scaffold 4, 5, and 2; IBT 40293: scaffolds 1203, 1206, and 386; IBT 7711: scaffolds 1035, 142, 1199, and 500) (compare Figure 6).

Strain ID	AC2		Downstream of AC2	
	Mean Identity (%)	Mean Coverage (%)	Mean Identity (%)	Mean Cover (%)
51-11	88.4	100	74.5	86.0
IBT 40293	89.2	98.8	54.1	64.4
IBT 7711	88.7	100	73.5	84.2

## Data Availability

Not applicable.

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
