# Peer review of "The Evolution of the Satratoxin and Atranone Gene Clusters of *Stachybotrys chartarum"

_jof, 2022, doi:10.3390/jof8040340_

Round 1

Reviewer 1 Report

Authors provide the first model for the S. chartarum genotypes. This is a fundamental work for the prevention of health diseases due to the S. chartarum secondary metabolites, stratoxin and atranone. The paper is well written and the scientific aspects are well discussed. Personally, I find the work very interesting and full of information for, for example, the development of new drugs targeting some of the genes described.

Author Response

Authors provide the first model for the S. chartarum genotypes. This is a fundamental work for the prevention of health diseases due to the S. chartarum secondary metabolites, stratoxin and atranone. The paper is well written and the scientific aspects are well discussed. Personally, I find the work very interesting and full of information for, for example, the development of new drugs targeting some of the genes described.

A: We thank Reviewer 1 for these kind comments.

Reviewer 2 Report

Manuscript Title: The evolution of the satratoxin and atranone gene clusters of Stachybotrys chartarum 

Authors: Sebastian Ulrich, Katharina Lang , Ludwig Niessen , Christiane Baschien , Robert Kosicki , Magdalena Twarużek , Reinhard Karl Straubinger , Frank Ebel

Manuscript ID: jof-1621592

General comment:

In the manuscript, the authors analyzed the scaffolds harboring the satratoxin gene clusters SC1-3 and the atranone gene clusters AC1 and 2. Additionally, the authors performed a deep re-sequencing of these gene clusters and an LC-MS/MS analysis of the atranone production using S. chartarum strains IBT 40293 and IBT 40288. This study provides several clues for a better understanding of the evolutionary processes that shaped the present genotypes and their gene clusters and suggest S. chartarum as a promising model organism for further research on the evolution of mycotoxin-specific gene clusters.

In general, this paper is clearly laid out, well planed and easy to read. The experiments are well designed. Some specific suggestions or questions are listed below:

1. Abstract:LC-MS/MS, please use full name for the first time.

2. Introduction: In this study, we have analyzed the scaffolds harboring the satratoxin gene clusters SC1-3 and the atranone gene clusters AC1 and 2 (Table S1). Please remove the result (Table S1)to the Results

3. Introduction: line 84-86, please revised the sentence as “Additionally, the authors performed a deep re-sequencing of these gene clusters and an LC-MS/MS analysis of the atranone production using chartarumstrains IBT 40293 and IBT 40288.”

4. Introduction: The Introductionsection should focus on the research progress related to the topic and emphasize the innovation of this research. However, the novelty and significance of the topic were not highlighted, please modify the introduction more clearly.

5. Conclusions: please revise “Stachybotrys chartarum”as “S. chartarum”. Please ensure that abbreviations/acronyms are defined the first time they appear in each of three sections: the abstract; the main text; the first figure or table.

6. The Referencessection is not clear. Please revise it. In addition, many of the references have been superceded and more modern ones are required, such as Journal of the South African Veterinary Association 1979, 50, 73-81.; Experimental Mycology 1982, 6, 25-30; Journal of antibiotics (Tokyo) 1982, 35, 875-881, etc.

Reviewer 3 Report

The paper describes an extensive study on two gene clusters from Stachybotrys chartarum, responsible for the biosynthesis of satratoxin and atranone. The manuscript is well-written and provides new information that is of possible interest to international readership. I have only minor comments and suggestions on how to improve the paper, which were given below.

Introduction is logical and concise. It gives enough information to lead into the subject. I do not have any comments on this section.

Materials and Methods are very detailed and accurate. The experiments can easily be repeated based on these data.

My main concern relates to the Results section. In my opinion, it is slightly oversized and could be somehow shortened, particularly for there are some paragraphs that interpret the results and, thus, belong rather to the Discussion, and also there are some repeated information in Results and Discussion chapters. Moreover, there is a mistake in line 602 (F. graminearum does not produce T-2 toxin, which is Group A trichothecene). Generally, the presentation of the results is clear and convincing. I’ve got a feeling that the terms ‘chemotype’ and ‘genotype’ are used interchangeably. I think that in this context ‘chemotype’ should be used. I also miss the confirmation of the ability to produce satratoxin and atranone with respective strains by the analysis of metabolites’ levels.

Discussion: avoid repeating the information already present in the Results. Some statements, especially dealing with proposed scenarios for chemotype evolutionary history are pretty speculative. I would suggest to minimize this and/or support with more reference data for different clusters and species.

Author Response

My main concern relates to the Results section. In my opinion, it is slightly oversized and could be somehow shortened, particularly for there are some paragraphs that interpret the results and, thus, belong rather to the Discussion, and also there are some repeated information in Results and Discussion chapters 1. Moreover, there is a mistake in line 602 (F. graminearum does not produce T-2 toxin, which is Group A trichothecene).2 Generally, the presentation of the results is clear and convincing. I’ve got a feeling that the terms ‘chemotype’ and ‘genotype’ are used interchangeably.3 I think that in this context ‘chemotype’ should be used. I also miss the confirmation of the ability to produce satratoxin and atranone with respective strains by the analysis of metabolites’ levels.4

A:

1: We have shortened the Discussion section and also moved some sentences from Results to the Discussion (L239; 284; 504; L227; 237; 302; 318; 342; 381; 428; 670; 690; 736).

2: We have corrected this mistake and are grateful for this advice (L602).

3: This study is largely focused on genetic information. We used short forms like ‘A-type’ for strains belonging to genotype A. This is now explicitly mentioned in the text ("In the following text we refer to individual strains as A- or S-type in accordance to their genotype.").

4: We also would have preferred to measure the metabolites for all four strains for which genetic information is present in the data base, unfortunately, only two strains were available. We have tried to obtain the other strains, but without success.

Discussion: avoid repeating the information already present in the Results. Some statements

1, especially dealing with proposed scenarios for chemotype evolutionary history are pretty speculative. I would suggest to minimize this and/or support with more reference data for different clusters and species.2

A:

1: We have tried to remove such repetitions from the revised manuscript.

2: The evolution of fungal gene clusters is a field that is still in its infancy and hard evidence is rare. We found genetic evidence that suggests certain evolutionary events, but we agree with Reviewer 2 that any conclusion or any model will in part be speculative. The events described in the text could explain our data, but we don’t claim that we provide an ultimate truth. We believe however that the presentation of a model can be valuable and fruitful.

Reviewer 4 Report

The manuscript explains “The evolution of the satratoxin and atranone gene clusters of Stachybotrys chartarum”The findings are quite satisfactory though some minor corrections required before the acceptance. 

  1. The degree of precision of "Material and method" is relatively moderate and need to improve. Indicate how much mycelium was added (line 148).
  2. Make a schematic diagram of the performed work it will ease to understand the work to readers.
  3. Discussion is too lengthy and need to improve in the light of the current findings.
  4. Follow the same format for the units suggested by the journal e.g ul (line 123, 124, 127) or uL (line 160) and ml (line127) or mL (line 134, 148).

Author Response

The manuscript explains “The evolution of the satratoxin and atranone gene clusters of Stachybotrys chartarum”. The findings are quite satisfactory though some minor corrections required before the acceptance.

Thank you very much for improving our manuscript by your suggestions. The reviewer can find all changes in the manuscript in marker mode.

  1. The degree of precision of "Material and method" is relatively moderate and need to improve. Indicate how much mycelium was added (line 148).

A: We have added the approx. amount of mycelium (approx. 1.2 g/wet weight) (L154).

  1. Make a schematic diagram of the performed work it will ease to understand the work to readers.

A: We have added a schematic diagram as a supplementary figure (Figure S1) to material and methods (L83).

  1. Discussion is too lengthy and need to improve in the light of the current findings.

A: We agree with you on the fact that some repetitive elements were present and the discussion can be shortened. Accordingly, we have shortened and moved several sentences. We have moved sentences from the results part into the discussion (L239; 284; 504). We also deleted repetitive parts to shorten the results and discussion part as suggested (L227; 237; 302; 318; 342; 381; 428; 670; 690; 736). Furthermore, we have once again checked the text for english language and style and improved the manuscript.

  1. Follow the same format for the units suggested by the journal e.g ul (line 123, 124, 127) or uL (line 160) and ml (line127) or mL (line 134, 148, 188).

A: We have adapted the format as suggested (L123; 127; 128) and according to the journal author guidelines.